# A model of preferential pairing between epithelial and dendritic cells in thymic antigen transfer

Matouš Vobořil[1†], Jiří Březina[1,2†], Tomáš Brabec[1], Jan Dobeš[1,2], Ondřej Ballek[1], Martina Dobešová[1], Jasper Manning[1], Richard S Blumberg[3], Dominik Filipp[1*]

[1]Laboratory of Immunobiology, Institute of Molecular Genetics of the Czech Academy of Sciences, Prague, Czech Republic; [2]Department of Cell Biology, Charles University, Faculty of Science, Charles University, Prague, Czech Republic; [3]Division of Gastroenterology, Hepatology, and Endoscopy, Department of Medicine, Brigham and Women's Hospital, Harvard Medical School, Boston, United States

**Abstract** Medullary thymic epithelial cells (mTECs), which produce and present self-antigens, are essential for the establishment of central tolerance. Since mTEC numbers are limited, their function is complemented by thymic dendritic cells (DCs), which transfer mTEC-produced self-antigens via cooperative antigen transfer (CAT). While CAT is required for effective T cell selection, many aspects remain enigmatic. Given the recently described heterogeneity of mTECs and DCs, it is unclear whether the antigen acquisition from a particular TEC subset is mediated by preferential pairing with a specific subset of DCs. Using several relevant *Cre*-based mouse models that control for the expression of fluorescent proteins, we have found that, in regards to CAT, each subset of thymic DCs preferentially targets a distinct mTEC subset(s). Importantly, XCR1[+]-activated DC subset represented the most potent subset in CAT. Interestingly, thymic DCs can also acquire antigens from more than one mTEC, and of these, monocyte-derived dendritic cells (moDCs) were determined to be the most efficient. moDCs also represented the most potent DC subset in the acquisition of antigen from other DCs. These findings suggest a preferential pairing model for the distribution of mTEC-derived antigens among distinct populations of thymic DCs.

**\*For correspondence:**
dominik.filipp@img.cas.cz

[†]These authors contributed equally to this work

**Competing interest:** The authors declare that no competing interests exist.

## Editor's evaluation

This manuscript will be of interest to immunologists studying mechanisms of thymic central tolerance. The study elegantly makes use of multiple genetic mouse models to generate data supporting the conclusion that different dendritic cell subsets in the thymus capture self-antigens from distinct subsets of thymic epithelial cells.

## Introduction

Central tolerance, which operates during T cell development in the thymus, can result in the elimination of self-reactive T cells or deviation to a thymic regulatory T cell (tTreg) lineage (*Klein et al., 2019*). The underlying principle of this event compels immature T cells to test their T cell receptor (TCR) for potential self-reactivity through the scanning of self-antigens that are presented by antigen-presenting cells (APCs). Among all thymic APCs, thymic epithelial cells (TECs) are central to this selection process (*Klein et al., 2014*). Based on their localization within the thymus, TECs are generally divided into two major populations: cortical thymic epithelial cells (cTECs) and medullary thymic epithelial cells (mTECs) (*Derbinski et al., 2001*). Recently, results of single-cell RNA sequencing (scRNAseq) revealed

an unexpected heterogeneity of mTECs with at least five distinct subsets defined by developmental stage, transcription profile, and function (referred to as mTEC-I, -II, -IIIa, IIIb, and Tuft cells) (*Baran-Gale et al., 2020*; *Bornstein et al., 2018*; *Miller et al., 2018*).

Due to their unique ability to express and present more than 80% of the protein-coding genome, mTECs are well-adapted to serve as a principal self-antigen-producing cellular component of central tolerance (*Brennecke et al., 2015*; *Meredith et al., 2015*; *Sansom et al., 2014*). This is in part facilitated by the expression of the autoimmune regulator (Aire). Aire controls the gene expression of a large set of tissue-restricted antigens (TRAs) that are found only in the immune periphery (*Derbinski et al., 2001*). Interestingly, an effective display of a complete set of thymically expressed TRAs is achieved by their combinatorial mosaic expression by each mTEC with any particular TRA expressed by only 1–3% of mTECs (*Derbinski et al., 2008*) while a single mTEC is capable of expressing up to 300 different TRAs (*Meredith et al., 2015*; *Sansom et al., 2014*). However, mTEC subsets are not equal in terms of Aire expression and TRA presentation. During their progression through mTEC-I, -II, -IIIa, and -IIIb stages, the highest Aire and TRA expression is observed within the mTEC-II stage, historically referred to as mTEC$^{high}$. As mTEC-II cells enter pre-post Aire and post-Aire phases (phase -IIIa and -IIIb, respectively), they downregulate the expression of Aire, although their TRA protein levels remain high, making them available for transfer to other cells (*Kadouri et al., 2020*). It is noteworthy that the expression of TRA in mTEC-I (also referred to as mTEC$^{low}$) is limited (*Baran-Gale et al., 2020*; *Bornstein et al., 2018*).

The relatively low number of mTECs in comparison to the sheer number of developing T cells, coupled to mosaic and stage-restricted expression of TRAs, places significant constraints on the process of T cell selection. To overcome this limitation, TRAs from apoptotic mTECs can be transferred into, and indirectly presented to, T cells by thymic dendritic cells (DCs) via the process of cooperative antigen transfer (CAT) (*Gallegos and Bevan, 2004*; *Koble and Kyewski, 2009*). It has been demonstrated that CAT is critical for the establishment of central tolerance to mTEC-derived self-antigens (*Lancaster et al., 2019*; *Perry et al., 2014*; *Perry et al., 2018*). Despite its importance, the elucidation of the basic principles of CAT has been hampered by the complexity of thymic DC populations.

In general, thymic DCs can be divided into two major categories: plasmacytoid dendritic cells (pDCs) and classical dendritic cells (cDCs), the latter of which can be subdivided into cDC1 and cDC2 subsets (*Guilliams et al., 2014*). Previous studies have shown that these DC subsets vary in their capacity to acquire mTEC-derived antigens (*Kroger et al., 2017*; *Lancaster et al., 2019*; *Voboøil et al., 2020*). The cDC1 subset has been shown to strongly acquire GFP antigen from mTEC in an *Aire-GFP* mouse model (*Perry et al., 2018*) while the cDC2 subset robustly acquired mOVA antigen in a *RIP-mOVA* mouse model (*Lancaster et al., 2019*). Since the expression of Aire-driven GFP and mOVA in the thymus was largely restricted to Aire$^+$ mTECs (*Gardner et al., 2008*) and mTEC$^{Low}$/post-Aire mTECs, respectively (*Mouri et al., 2017*), it has been inferred that distinct subsets of thymic DCs acquire antigens from distinct subsets of mTECs. However, our recent scRNAseq analysis, along with data from the human thymus cell atlas study, exposed a much broader heterogeneity of DCs in the thymus of mice and humans (*Park et al., 2020*; *Voboøil et al., 2020*). Thus, a more comprehensive analysis is needed to determine the mode of CAT between defined subsets of TECs and DCs as well as other means of thymic antigen spreading.

In this study, we used several *Cre* reporter mouse models in which the expression of fluorescent TdTOMATO protein (TdTOM) is restricted to different subsets of TECs. We present evidence that suggests that distinct subsets of thymic DCs preferentially acquire TdTOM from a specific TEC subset. Using the *Confetti*$^{Brainbow2.1}$ system, we have also shown that CAT can occur as a repetitive event whereby a single thymic CD11c$^+$ cell can acquire antigen from two or more individual TECs. Furthermore, based on our data, we postulate that antigen transfer can also occur between DC subsets themselves. Thus, this dataset suggests a deterministic model of preferential engagement of specific mTEC and DC subsets for directional thymic antigen distribution.

## Results
### Thymic epithelial cell models of cooperative antigen transfer
In recent years, the robustness of scRNAseq has yielded a vast amount of information regarding the thymic APCs inventory and has been instrumental in revealing a list of suitable markers (*Baran-Gale*

*et al., 2020*; *Bautista et al., 2021*; *Bornstein et al., 2018*; *Dhalla et al., 2020*; *Park et al., 2020*; *Voboříl et al., 2020*; *Wells et al., 2020*). The combinatorial specificity of these markers has led us to design novel flow cytometry gating strategies that allow for the study of CAT.

To understand antigen transfer trajectories within the intricate network of all subsets of TECs and CD11c$^+$ APCs identified thus far, we first established mouse models where cytoplasmic expression of TdTOM is preferentially confined to distinct TEC subsets. By crossing three previously characterized *Cre*-based mouse models with a *Rosa26$^{TdTOM}$* mouse strain, we generated (i) *Foxn1$^{Cre}$Rosa26$^{TdTOM}$* (*Foxn1$^{Cre}$*) mice that express TdTOM in all populations of CD45$^-$EpCAM$^+$ TECs (*Gordon et al., 2007*; *Voboříl et al., 2020*), (ii) *Csnb$^{Cre}$Rosa26$^{TdTOM}$* (*Csnb$^{Cre}$*) with *Casein β* (*Csnb*) loci operating as an Aire-independent TRA that confines TdTOM expression to the mTEC$^{High}$ subset and their closest progenitors and progeny (*Bornstein et al., 2018*; *Tykocinski et al., 2010*), and (iii) *Defa6$^{iCre}$Rosa26$^{TdTOM}$* (*Defa6$^{iCre}$*). The latter model represents the 'classical' Aire-dependent TRA model, in which TdTOM is expressed in 1–3% of Aire$^+$ mTEC$^{High}$ cells as well as post-Aire mTEC progeny (*Adolph et al., 2013*; *Dobeš et al., 2015*; *Figure 1a–c*). A quantitatively distinct representation of TdTOM$^+$ cells in these three mouse models (*Figure 1e*) has been corroborated by microscopic examination of thymic sections (*Figure 1d*).

The gating strategy implemented to assess the frequency of TdTOM-labeled CD45$^-$EpCAM$^+$ TEC subsets in the *Cre* models introduced above is shown in *Figure 1—figure supplement 1a*. Six subsets of TECs were distinguished: cTEC, mTEC$^{Low}$, mTEC$^{High}$, two subsets of LY6D$^+$ terminally differentiated subsets: pre-post Aire and post-Aire mTECs, and L1CAM$^+$ thymic Tuft cells. To show differences in the preferential expression of TdTOM among TEC subsets, we first determined the percentage representation of TEC subsets within the TdTOM$^+$ gate as well as each TEC subset overall (an example of gating is illustrated in *Figure 1—figure supplement 1b*, top and bottom rows, respectively) in all three mouse models used. While the frequencies of TEC subsets overall showed comparable values among the three models (all TECs), those from the TdTOM$^+$ gate (TdTOM$^+$ TECs) differed (*Figure 1—figure supplement 1c*, blue and red squares, respectively). These differences are readily apparent when representative samples from each of the reporter model used are compared (*Figure 1f*). The ratio between the percentage of a given TEC subset from TdTOM$^+$ gate normalized to the percentage of the same TEC subset overall indicates the relative over- or underrepresentation of the former within the thymic CD45$^-$EpCAM$^+$TdTOM$^+$ population among the three models used (*Figure 1g*). Thus, whereas the cTEC and mTEC$^{Low}$ subsets were increased in *Foxn1$^{Cre}$* and the mTEC$^{High}$ subset in the *Csnb$^{Cre}$* model, the ratios of pre-post Aire and Tuft mTECs were increased in the *Defa6$^{iCre}$* model (*Figure 1g*) and post-Aire mTECs were diminished in the *Csnb$^{Cre}$* model. A comparative analysis of the percentage of TdTOM$^+$ cells within the parent TEC population confirmed that whereas the frequency of TdTOM$^+$ cells in the *Foxn1$^{Cre}$* model reached as expected nearly 100%, it dramatically decreased in the *Csnb$^{Cre}$* and to a greater extent in the *Defa6$^{iCre}$* model (*Figure 1—figure supplement 1d*). It is also clear that the major contribution to the TdTOM$^+$ pool of cells in the *Csnb$^{Cre}$* and *Defa6$^{iCre}$* models came from the mTEC$^{High}$ and pre-post Aire subsets (*Figure 1—figure supplement 1d*). This data validated the utility of the *Cre*-based *Rosa26$^{TdTOM}$* mouse models for the study of CAT since in each model the expression of TdTOM protein was predictably and reproducibly enriched in different subsets of TECs.

## Antigen transfer of TdTOM to thymic dendritic cells

Having characterized the distinct distribution of TdTOM in TEC subsets in our *Cre*-based *Rosa26$^{TdTOM}$* mouse models, we next tested the distribution of TdTOM among its acceptors, the thymic population of CD11c$^+$ cells (*Figure 2a*). As previously shown (*Voboříl et al., 2020*) as well as in *Figure 2b and c*, TdTOM is mostly acquired by CD11c$^+$ cells. The robustness of this transfer, which is heavily dependent on the type of *Cre*-based *Rosa26$^{TdTOM}$* mouse model, was then examined. Whereas TdTOM positivity was observed in ~6% of CD11c$^+$ cells in the *Foxn1$^{Cre}$* model, its frequency in *Csnb$^{Cre}$* and *Defa6$^{iCre}$* was limited to ~0.6% and ~0.02%, respectively (*Figure 2c*). Interestingly, since the frequency of TdTOM$^+$ TECs was analogously decreased across the *Foxn1$^{Cre}$*, *Csnb$^{Cre}$*, and *Defa6$^{iCre}$* mouse models (*Figure 1c*), the ratio between the frequency of TdTOM$^+$CD11c$^+$ and TdTOM$^+$CD45$^-$EpCAM$^+$ TECs was comparable (*Figure 2d*). These analyses argue for the similarity in CAT efficiency between donor TECs and CD11c$^+$ APC acceptors, irrespective of the robustness and cell-subset range of TdTOM expression in TECs.

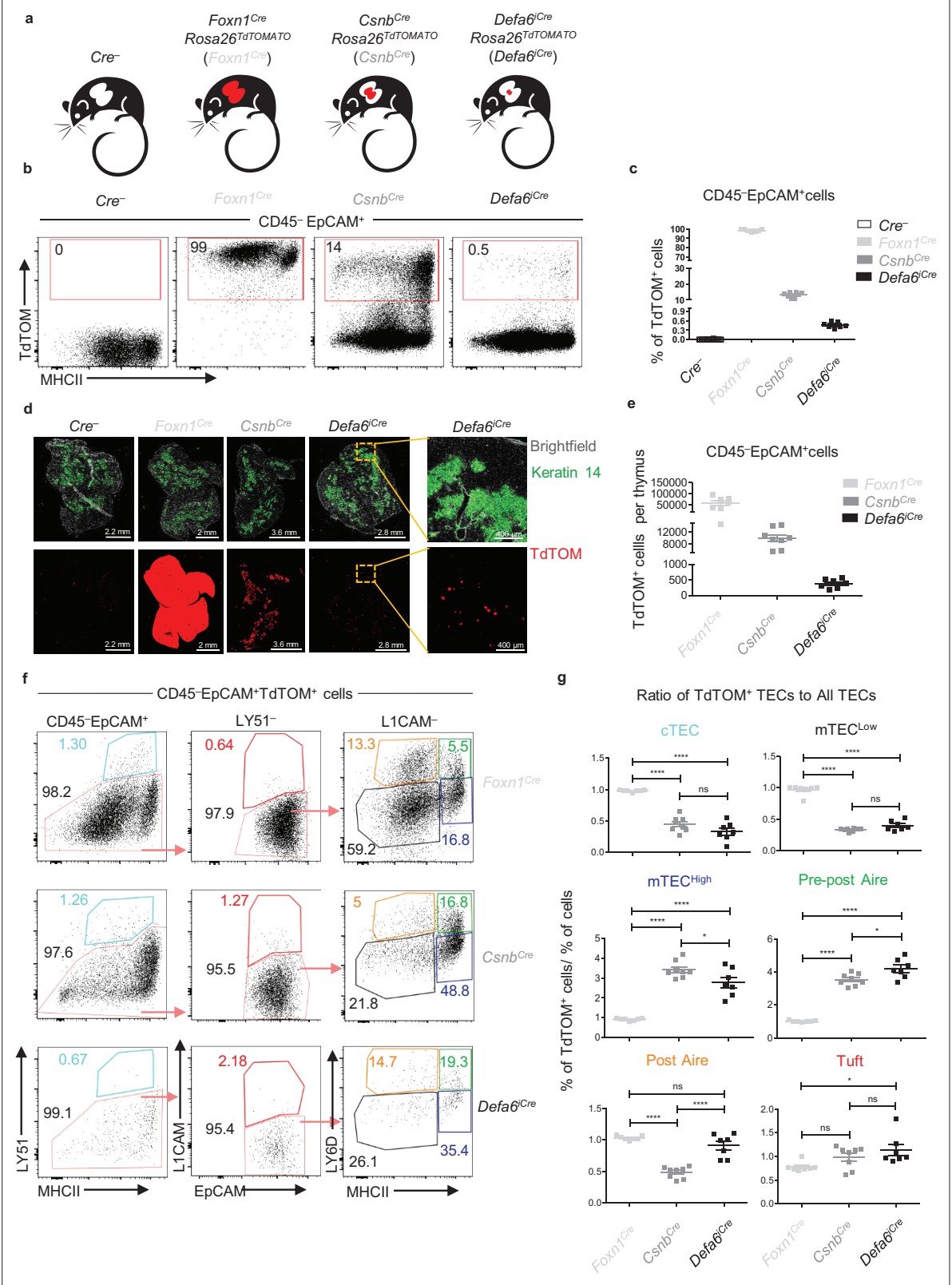

**Figure 1.** The phenotype and frequency of thymic epithelial cell (TEC) subsets in *Cre*-based mouse models of cooperative antigen transfer (CAT). (**a**) Mouse models of CAT with confined expression of TdTOM to distinct TEC subsets. (**b**) Representative flow cytometry plots showing the frequency of TdTOM⁺ cells among MACS-enriched CD45⁻EpCAM⁺ cells isolated from *Foxn1^Cre^Rosa26^TdTOM^* (*Foxn1^Cre^*), *Csnb^Cre^Rosa26^TdTOM^* (*Csnb^Cre^*), and *Defa6^iCre^Rosa26^TdTOM^* (*Defa6^iCre^*) mice. (**c**) Quantification of TdTOM⁺ cells from *Figure 1b* (mean ± SEM, n = 7–12 mice from three independent

*Figure 1 continued on next page*

*Figure 1 continued*

experiments). (**d**) Microscopy images of thymic sections from *Foxn1^Cre^*, *Csnb^Cre^*, *Defa6^iCre^*, and *Cre⁻* control mice depicting the levels of TdTOM expression and its restriction to keratin 14⁺ areas demarcating thymic medulla. Yellow dashed lines denote the zones that are shown in higher magnification. Scale bars and color code are provided. (**e**) Quantification of the absolute numbers of CD45⁻EpCAM⁺TdTOM⁺ cells isolated from *Foxn1^Cre^*, *Csnb^Cre^*, and *Defa6^iCre^* thymi counted by flow cytometry. (**f**) Representative flow cytometry plots of different TEC subsets in *Foxn1^Cre^*, *Csnb^Cre^*, and *Defa6^iCre^* mice. (**g**) Quantification of the ratios of the frequencies of TEC subsets within CD45⁻EpCAM⁺TdTOM⁺ cells to the frequencies of TEC subsets within CD45⁻EpCAM⁺ cells in *Foxn1^Cre^*, *Csnb^Cre^*, and *Defa6^iCre^* mice (mean ± SEM, n = 7–8 mice from three independent experiments). The ratios were calculated using percentages in *Figure 1—figure supplement 1c*. Statistical analysis in (**g**) was performed using one-way ANOVA with Bonferroni´s multiple comparison test, $*p \leq 0.05$, $**p \leq 0.01$, $***p \leq 0.001$, $****p < 0.0001$, ns, not significant.

The online version of this article includes the following source data and figure supplement(s) for figure 1:

**Source data 1.** The phenotype and frequency of thymic epithelial cell (TEC) subsets in Cre-based mouse models of cooperative antigen transfer (CAT).

**Figure supplement 1.** Thymic epithelial cell (TEC) populations.

**Figure supplement 1—source data 1.** Thymic epithelial cell (TEC) populations.

To study CAT in the mouse models defined above, we determined seven subpopulations of thymic CD11c⁺ cells (*Figure 2—figure supplement 1a*). These cells comprise three major categories: B220⁺ pDCs, CD11c^Low^MHCII^Low^CX3CR1⁺ macrophage-like population (Mac), and CD11c⁺MHCII^High^ cells, which represent a conventional type of thymic DCs that have been subdivided into two groups, cDC1 and cDC2, defined by the expression of the chemokine receptors, XCR1 and SIRPα, respectively (*Li et al., 2009*). Recently, the SIRPα⁺ DCs were described to encompass a minimum of two different subpopulations, defined by the expression of MGL2 (CD301b) and CD14 to MGL2⁺CD14⁻ cDC2 and MGL2⁺CD14⁺ monocyte-derived dendritic cells (moDC) (*Vobořil et al., 2020*). It has also become evident that DCs could be phenotypically and functionally defined by their activation status (*Ardouin et al., 2016*; *Oh et al., 2018*; *Park et al., 2020*; *Vobořil et al., 2020*). Hence, two phenotypically distinct subsets of activated dendritic cells (aDCs), CCR7⁺XCR1⁺ and CCR7⁺XCR1⁻, can be identified (*Figure 2—figure supplement 1a*). A comparative analysis of the ability of each of these thymic CD11c⁺ APC subsets to acquire TEC-derived TdTOM showed that XCR1⁺ aDCs were the most efficient cells involved in CAT irrespective of the *Cre*-based *Rosa26^TdTOM^* model used. On the other hand, while Macs and pDCs were relatively inefficient, the remaining subsets varied in this efficiency depending on the *Cre*-model analyzed (*Figure 2—figure supplement 1b*). Using bone marrow (BM) chimeras of sublethally irradiated mouse models (*Foxn1^Cre^*, *Csnb^Cre^*, and *Defa6^iCre^*) reconstituted with congenically marked BM cells isolated from WT animals, we verified that TdTOM is indeed transferred from TECs to all subpopulations of thymic CD11c⁺ APCs and is not endogenously expressed by these APCs themselves (*Figure 2—figure supplement 1c–f*).

Since the frequency of each CD11c⁺ APC subset as well as their capacity to acquire TEC-derived antigen differs, we next assessed their contribution to CAT in all three *Cre*-based *Rosa26^TdTOM^* mouse models. Due to the comparative nature of this approach (comparing the efficiency of CAT for each CD11c⁺ APC subset in each *Cre* model), we first performed an unsupervised flow cytometry analysis of all CD11c⁺TdTOM⁺ cells concatenated from 10 independent samples from each of the *Cre*-based mouse models, that is, 30 samples in total (*Figure 2e*). Based on the markers shown in *Figure 2—figure supplement 2a*, we identified all phenotypically distinct CD11c⁺ APC subsets in the resulting tSNE plot (*Figure 2—figure supplement 2a and b*). Analyzing each of the *Cre*-based *Rosa26^TdTOM^* mouse models individually (*Figure 2f*), the data revealed that whereas the contribution of cDC1s and moDCs to CAT was robust in all three models, the contribution of pDCs, Macs, cDC2s, and both populations of aDC subsets varied among the models. To investigate the preferential uptake of TdTOM among DC subsets, we compared the percentage representation of TdTOM⁺ DC subsets (TdTOM⁺ DCs) with the frequency of DC subsets among the entire DC population (All DCs) (*Figure 2—figure supplement 2c*) and calculated the ratios as similarly done in *Figure 1* (*Figure 2g*). Notably, the ratio between the percentage of TdTOM⁺ DC subset and the overall percentage for cDC2s, pDCs, and Macs was increased in the *Foxn1^Cre^* mouse model. In contrast, this ratio for XCR1⁺ and XCR1⁻ aDCs was the lowest in *Foxn1^Cre^*, with an increase detected in *Csnb^Cre^*, and the highest ratio detected in the *Defa6^iCre^* model (*Figure 2g*). Taken together, this data shows that the extent of the involvement of each DC subset in CAT depends on the distribution of TdTOM protein expression among the different subtypes of TECs and/or the overall proportion of TECs expressing the TdTOM. In this way,

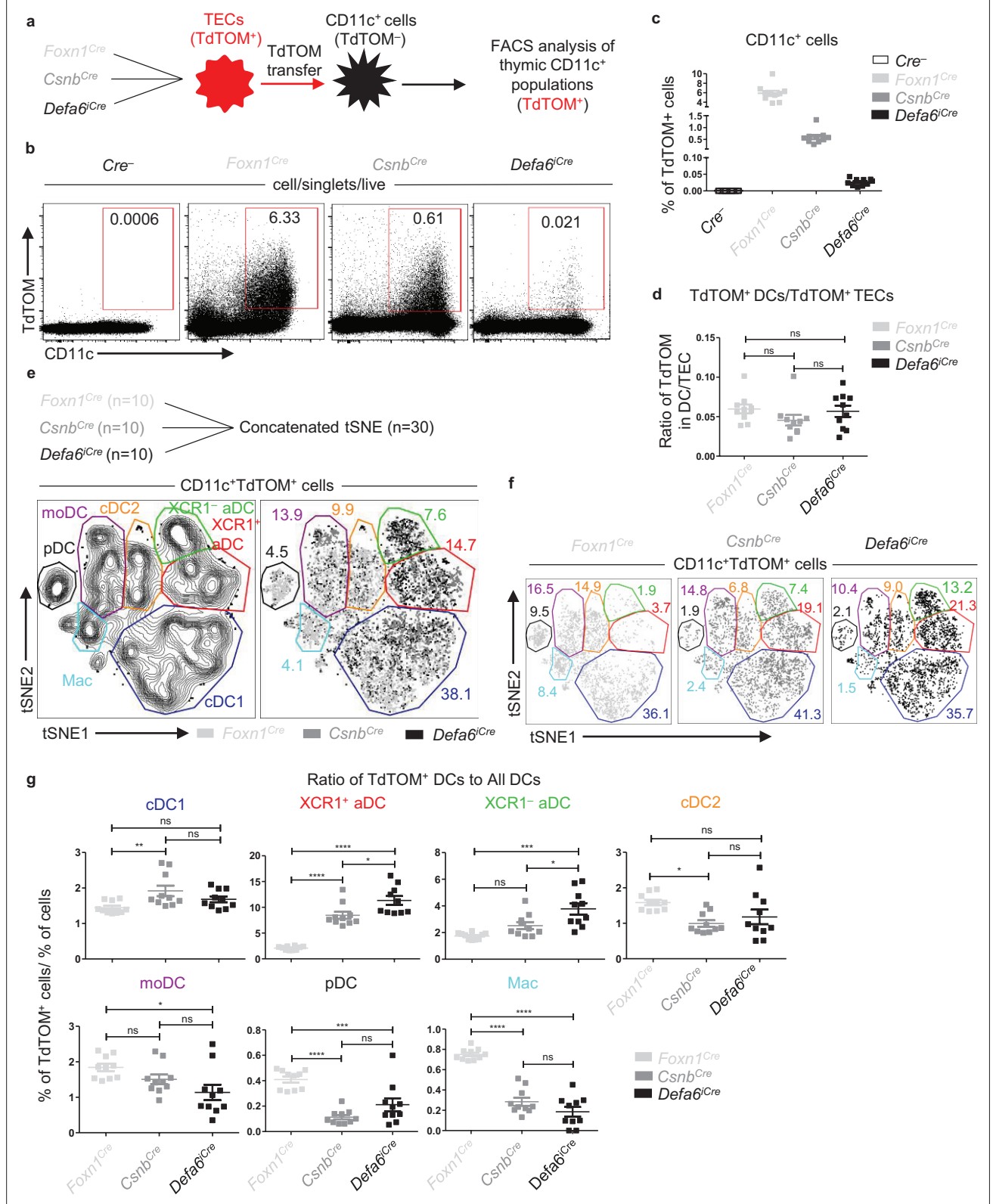

**Figure 2.** Antigen transfer of TdTOM to thymic dendritic cells (DCs). (**a**) Experimental design. (**b**) Representative flow cytometry plots comparing the frequency of TdTOM⁺CD11c⁺ cells among MACS-enriched CD11c⁺ thymic cells from mouse models described in (**a**). (**c**) Quantification of TdTOM⁺CD11c⁺ cells from (**b**) (mean ± SEM, n = 10 mice from a minimum of three independent experiments). (**d**) Comparison of the ratio between the frequency of TdTOM⁺CD11c⁺ (quantified in **c**) to TdTOM⁺ thymic epithelial cell (TEC) (quantified in *Figure 1c*) subsets in mouse models described in

*Figure 2 continued on next page*

*Figure 2 continued*

(**a**) (mean ± SEM, n = 10 mice from a minimum of three independent experiments). (**e**) Concatenated (n = 30 mice) and (**f**) separate (n = 10 mice) flow cytometry tSNE analysis of TdTOM⁺CD11c⁺ cells from the three mouse models described in (**a**). (**g**) Quantification of the ratios of the frequencies of DC subsets within CD11c⁺TdTOM⁺ cells to the frequencies of DC subsets within CD11c⁺ cells in *Foxn1Cre*, *CsnbCre*, and *Defa6iCre* mice (mean ± SEM, n = 10 mice from a minimum of three independent experiments). The ratios were calculated using percentages in **Figure 2—figure supplement 2c**. CD11c⁺ cell subsets were distinguished as in **Figure 2—figure supplement 1a**. Statistical analysis in (**d**) and (**g**) was performed using one-way ANOVA with Bonferroni´s multiple comparison test, *p≤0.05, **p≤0.01, ***p≤0.001, ****p<0.0001, ns, not significant.

The online version of this article includes the following source data and figure supplement(s) for figure 2:

**Source data 1.** Antigen transfer of TdTOM to thymic dendritic cells (DCs).

**Figure supplement 1.** Antigen transfer of TdTOMATO to thymic dendritic cells (DCs).

**Figure supplement 1—source data 1.** Antigen transfer of TdTOMATO to thymic dendritic cells (DCs).

**Figure supplement 2.** Thymic dendritic cell gating strategy defined by flow cytometry tSNE analysis.

**Figure supplement 2—source data 1.** Thymic dendritic cell gating strategy defined by flow cytometry tSNE analysis.

the assorted expression of TdTOM antigen by a limited but defined subset of TECs allows the visual identification of those DC subsets that engage these TEC subsets during CAT.

## Projecting preferential trajectories of CAT between TEC and thymic DC subsets

To reveal the possible combinations of TEC and DC subsets that are preferentially engaged in CAT, the ratio between the percentage of a given TEC subset from the TdTOM⁺ gate normalized to the percentage of the same TEC subset overall (**Figure 1g**), and the ratio between the percentages TdTOM⁺ cells and the entire population of a particular DC subset (**Figure 2g**) is depicted for each *Cre*-based mouse model (**Figure 3a**, upper and bottom rows, respectively). Upon inspection of these plots, a trend towards a decrease of mTECLow versus an increase of mTECHigh and pre-post Aire cells from *Foxn1Cre* to *CsnbCre* to *Defa6iCre* mouse models is apparent. On the other hand, a decrease in the ratio of cDC2s, pDCs, and Macs was observed while the contribution of XCR1⁺ and XCR1⁻ aDCs in the TdTOM⁺ gate was increased. The simplest interpretation of these correlations is the possibility of cDC2s, pDCs, and Macs preferentially acquiring antigens from the mTECLow subset, while the CAT to both populations of aDCs is likely associated with mTECHigh and pre-post Aire cells (**Figure 3a**).

To identify the predominant TEC-to-DC subsets trajectories of CAT, we performed a linear regression analysis of TEC and DC ratios across *Foxn1Cre*, *CsnbCre*, and *Defa6iCre* mouse models (**Figure 3a**). The data presented in **Figure 3b** confirmed the selectivity of each of the thymic DC subsets for certain TEC subset(s) from which they preferentially acquired antigens. Specifically, CAT to cDC1s significantly correlated with the expression of TdTOM in mTECHigh, XCR1⁺ aDCs with mTECHigh, pre-post Aire, and Tuft mTECs, whereas XCR1⁻ aDCs with mTECHigh and pre-post Aire mTECs. Interestingly, pDCs, and to a lesser extent Macs, correlated with antigen production in post-Aire mTECs. The positive correlation observed between pDCs and post-Aire mTECs is consistent with the fact that Hassall's corpuscles affect the migration and activation of thymic pDCs through the production of chemokines and cytokines (**Wang et al., 2019**). Macs, cDC2s, and pDCs were found to also correlate with the mTECLow population. In addition, the Macs population, and to a lesser extent cDC2s, significantly correlated with the expression of TdTOM in cTECs. This is consistent with the notion that both thymic Macs and cDC2s reside in the thymic cortex (**Baba et al., 2009**; **Breed et al., 2019**). It is also important to emphasize that CAT to pDCs, Macs, and cDC2s is highly affected by the frequency of total TdTOM⁺ TECs (**Figure 3c**). Thus, if the availability of TEC-derived antigens is limited, these subsets are outcompeted in CAT by other DC subsets.

Together, this data confirms the hypothesis that CAT occurs between subsets of TECs and thymic DCs in a selective manner, with the exception of moDCs, which failed to reveal a preference for any subset of TECs.

## Thymic moDCs are the most efficient subset in repetitive CAT

In experiments in which single-fluorescent protein (FP) transfer mouse models were utilized, most of the thymic CD11c⁺ subsets acquired antigens from more than one mTEC subset (**Figure 3c**). This poses the question of whether a single DC can take up antigens from several distinct TECs repetitively.

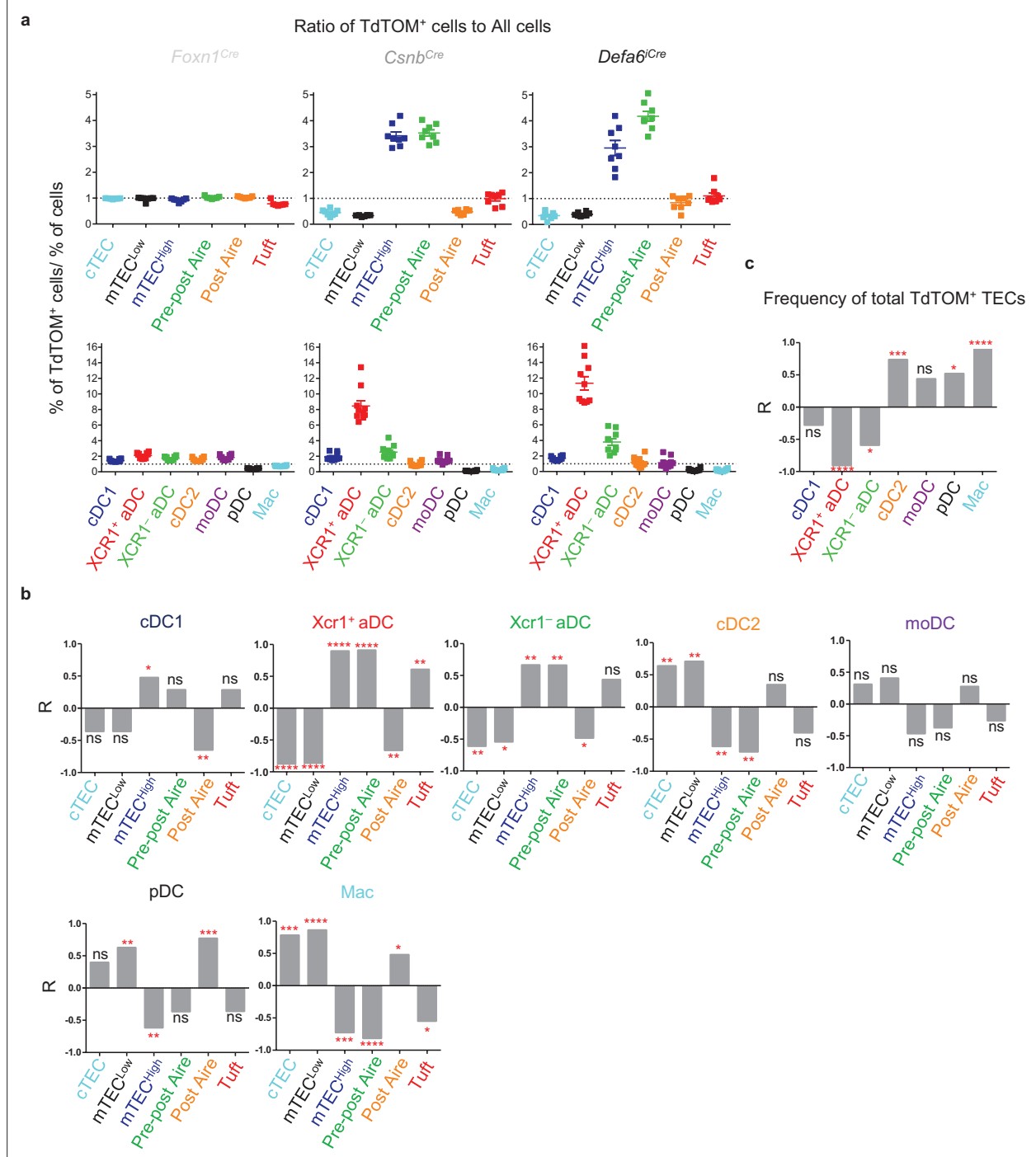

**Figure 3.** TdTOM antigen transfer to distinct thymic dendritic cell (DC) subsets correlates with its confined expression in phenotypically defined subsets of thymic epithelial cells (TECs). (**a**) Visualization of the ratios of TEC and DC subsets from *Figure 1g* (top) and *Figure 2g* (bottom), respectively, compared within *Foxn1Cre*, *CsnbCre*, and *Defa6iCre* models. Ratio = 1 is indicated by dotted line. (**b**) Bar graphs showing linear regression (R) between the ratios of TdTOM+ TECs and the indicated subset of TdTOM+ DCs from *Figures 1g and 2g* (n = 5–8 mice, from a minimum of three independent experiments). (**c**) Bar graph showing the R values between the ratio of TdTOM+ DCs from *Figure 2g* and the ratio of the frequency of TdTOM+ TECs (*Figure 1c*) to the frequency of all TECs (n = 8–10 mice from a minimum of three independent experiments). Statistical analysis in (**b**) and (**c**) was performed using a Pearson's product-moment correlation, *p≤0.05, **p≤0.01, ***p≤0.001, ****p<0.0001, ns, not significant.

The online version of this article includes the following source data for figure 3:

**Source data 1.** TdTOM antigen transfer to distinct thymic dendritic cell (DC) subsets correlates with its confined expression in phenotypically defined subsets of thymic epithelial cells (TECs).

To test this hypothesis, we utilized the *Foxn1^Cre^Confetti^Brainbow2.1^* mouse model in which cytosolic RFP, YFP, and membrane CFP are expressed individually or in combination by TECs. The transfer of these FPs to DCs (*Figure 4a*) was then measured. The expression of GFP, which should be present in the nucleus of *Foxn1^Cre^Confetti^Brainbow2.1^* TECs (*Snippert et al., 2010*), was recently reported to be abrogated (*Venables et al., 2019*). Visualizing TECs from *Foxn1^Cre^Confetti^Brainbow2.1^* and MHCII-EGFP mice, the latter used as a positive control, either separately or as a mixed cell suspension, confirmed that GFP is indeed absent in TECs isolated from *Foxn1^Cre^Confetti^Brainbow2.1^* mice (*Figure 4—figure supplement 1a*). Given that YFP and RFP/CFP are expressed from mutually exclusive cassettes in *Foxn1^Cre^Confetti^Brainbow2.1^* mice (*Snippert et al., 2010*), those TECs that express YFP do not express RFP and/or CFP and vice versa (*Figure 4—figure supplement 1b–d*). Therefore, those DCs that were positive for both RFP and YFP must have obtained these antigens from two or more distinct TECs (*Figure 4b*). We found that this multi-antigen transfer occurred nearly as frequently as the transfer from a single mTEC and that all CD11c^+^ APCs were involved in repetitive CAT. However, moDCs revealed the highest frequency of RFP^+^YFP^+^ cells, which suggests a high level of promiscuity in targeting TEC subsets (*Figure 4b*, *Figure 4—figure supplement 1e*).

The *Foxn1^Cre^Confetti^Brainbow2.1^* model also showed that the transfer of the CFP membrane antigen was observed less frequently than that of cytosolic antigens YFP and RFP. CFP transfer was largely mediated by XCR1^+^ aDCs, which exhibited more than a fivefold higher frequency of CFP positivity than any other CD11c^+^ subset (*Figure 4c*, *Figure 4—figure supplement 1f*). Among the CD11c^+^ cell subsets that acquired the CFP, we also analyzed the co-acquisition of the other two FPs (*Figure 4—figure supplement 1g*). As expected and consistent with their strong capacity to acquire FPs from more than one mTEC, the highest frequency of CFP^+^RFP^+^YFP^+^ cells was found in the moDC subset (*Figure 4—figure supplement 1g*, right plot). There were only a few CFP^+^YFP^+^ cells observed in the CD11c^+^ subsets, which correlated with the overall low abundance of CFP single positive mTECs (*Figure 4—figure supplement 1b*) and consequently a low probability of a sequential encounter with YFP^+^ and CFP single positive TEC by CD11c^+^ cells. Since XCR1^+^ aDCs were, in general, the most potent DC subset in CAT in *Foxn1^Cre^Confetti^Brainbow2.1^* mice, we imaged this subset with all possible FP^+^ variants using imaging flow cytometry (*Figure 4d*). It is of note that CFP was in direct contrast to other FPs that were localized mainly to the plasma membranes of CAT-experienced XCR1^+^ aDCs.

Taken together, using *Foxn1^Cre^Confetti^Brainbow2.1^* mice, we demonstrated that a single CD11c^+^ APC frequently acquired antigens from more than one mTEC and that the most potent subset in this repetitive CAT were moDCs. Moreover, we also showed that XCR1^+^ aDCs were effective in the acquisition of both cytosolic and membrane-bound antigens.

## Thymic CD11c^+^ cells can share their antigens

Apart from the other CD11c^+^ APCs analyzed, the moDC subset showed no specific preference for any TEC subset in CAT (*Figure 3b*). This, together with their highest capacity among other CD11c^+^ subsets for repetitive CAT (*Figure 4b*), led us to test their possible involvement in the acquisition of antigens from other thymic CD11c^+^ cells. We performed a mixed BM chimera experiment in which irradiated CD45.1^+^CD45.2^+^ WT mice were reconstituted with a mix of BM (50:50) isolated from CD45.1^+^ WT and CD45.2^+^ *Cd11c^Cre^Rosa26^TdTOM^* mice (*Figure 5a*, *Figure 5—figure supplement 1a*). Flow cytometric analysis showed that out of all CD45.1^+^CD11c^+^ cells, approximately 0,75% acquired TdTOM from CD45.2^+^CD11c^+^ cells (*Figure 5b and c*). While the contribution of both aDC subsets and cDC2s to CAT was robust, the highest frequency of TdTOM^+^ cells was found among the moDC subset (*Figure 5d*, *Figure 5—figure supplement 1b*). Thus, thymic CD11c^+^ cells, especially moDCs, acquired antigens not only from TECs but from other CD11c^+^ cells as well.

Together, this data demonstrates that the acquisition of antigens by the thymic population of CD11c^+^ cells is not restricted to TEC subsets but is extended to their own CD11c^+^ cells. Remarkably, among all thymic DCs, moDCs were the most efficient in this special type of 'cannibalistic' CAT.

## Discussion

This study, which has been based on the initial observations of others (*Lancaster et al., 2019*; *Mouri et al., 2017*; *Perry et al., 2018*), confirmed that CAT, that is, TEC-to-DC antigen-spreading, is not a random process. Using these studies, along with reports concerning the heterogeneity of thymic

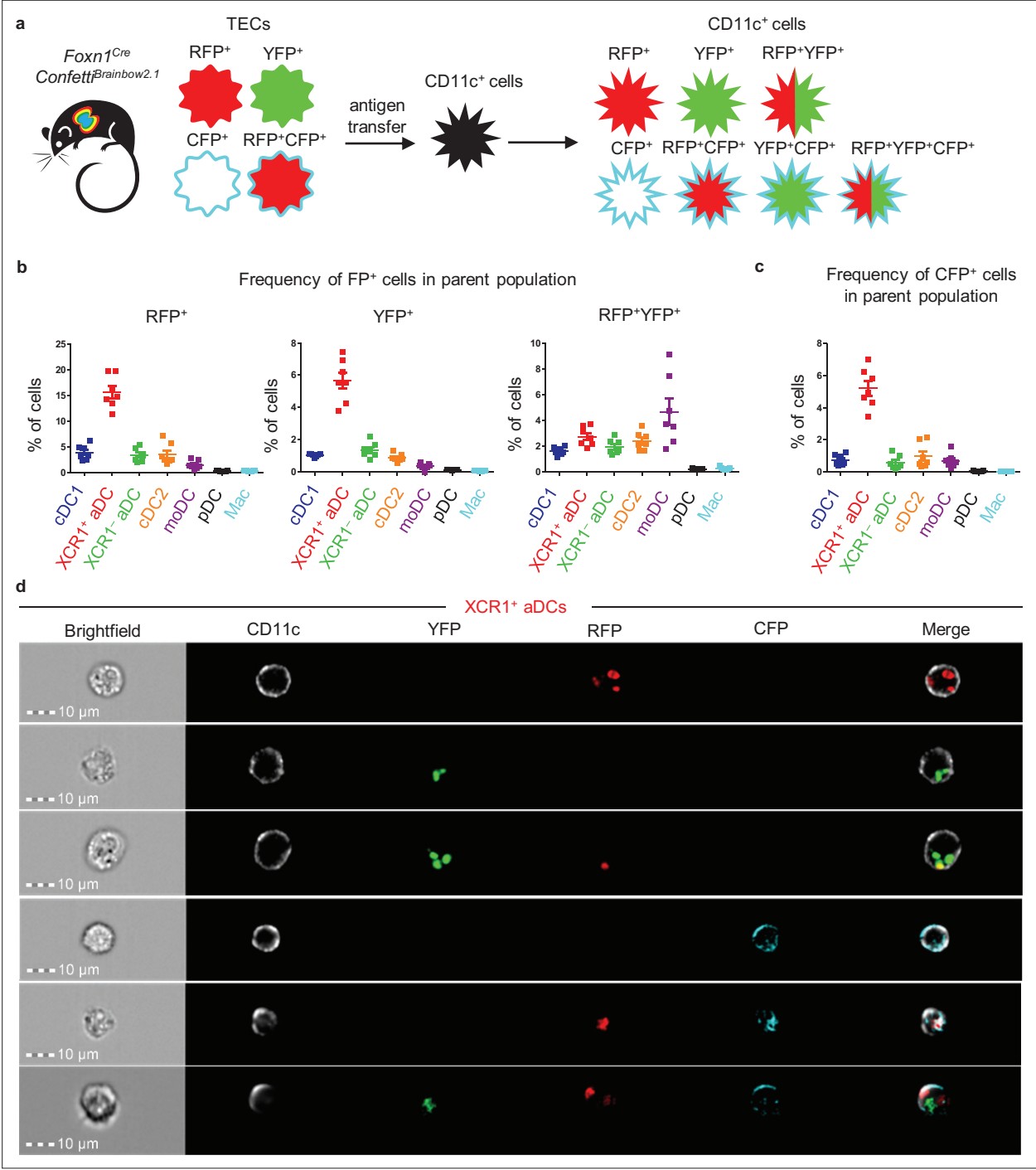

**Figure 4.** Thymic monocyte-derived dendritic cells (moDCs) efficiently acquire antigens from two or more thymic epithelial cell (TEC) in the *Foxn1*^CreConfetti^Brainbow2.1 mouse model. (**a**) Experimental design. (**b**) Quantification of the frequency of fluorescent protein+ (FP+) cells among the indicated dendritic cell (DC) subsets (mean ± SEM, n = 7 mice from three independent experiments). (**c**) Quantification of the frequency of CFP+ cells among the indicated DC subsets (mean ± SEM, n = 7 mice from three independent experiments). (**d**) Representative images from ImageStream analysis showing the localization of transferred FP in XCR1+ activated dendritic cell (aDC) from the thymus of *Foxn1*^CreConfetti^Brainbow2.1 (n = 2 independent experiments).

The online version of this article includes the following source data and figure supplement(s) for figure 4:

**Source data 1.** Thymic monocyte-derived dendritic cells (moDCs) efficiently acquire antigens from two or more thymic epithelial cell (TEC) cells in the Foxn1CreConfettiBrainbow2.1 mouse model.

**Figure supplement 1.** *Foxn1*^CreConfetti^Brainbow2.1 as a model of thymic cooperative antigen transfer.

**Figure supplement 1—source data 1.** Foxn1CreConfettiBrainbow2.1 as a model of thymic cooperative antigen transfer.

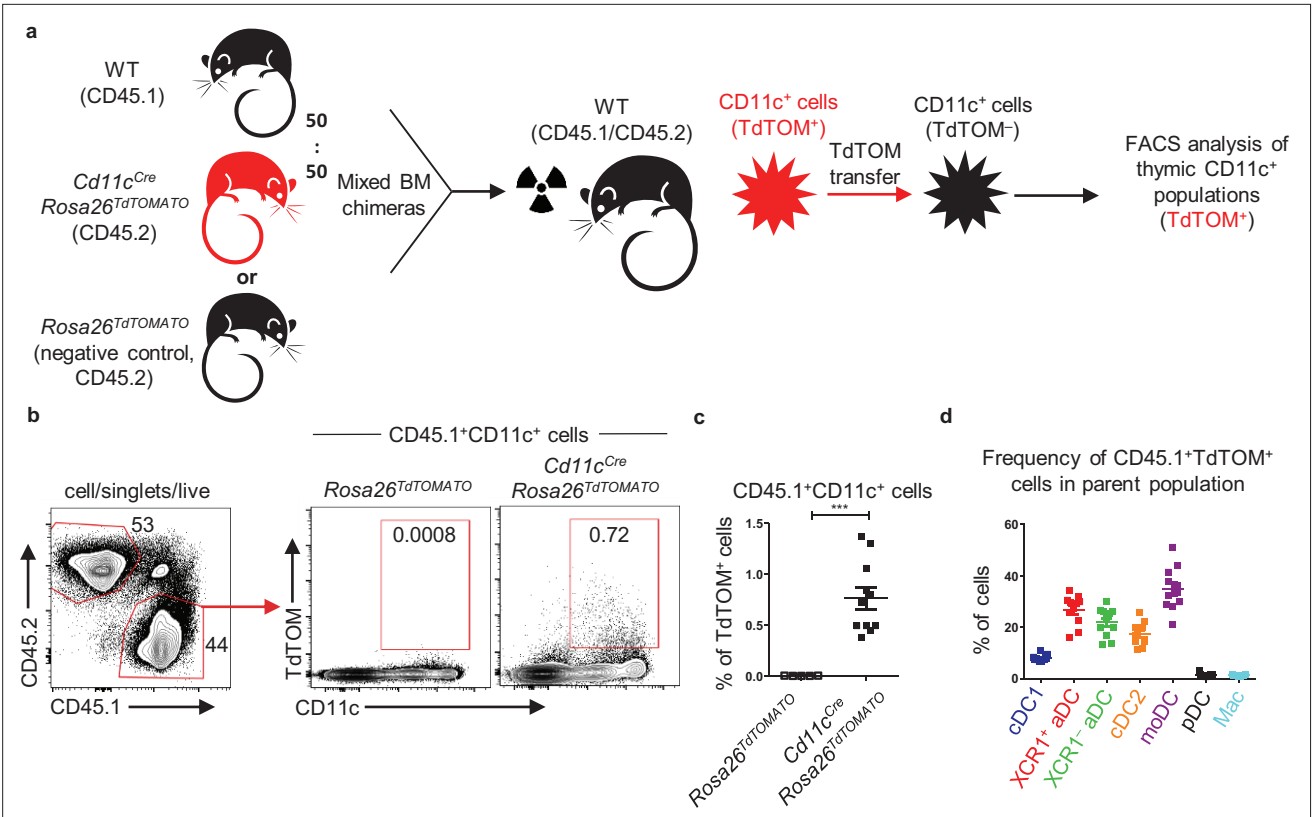

**Figure 5.** Thymic CD11c⁺ cells can share their antigens between each other. (**a**) Experimental design. (**b**) Representative flow cytometry plots showing the frequency of the transfer of TdTOM into CD45.1⁺CD11c⁺ cells among MACS-enriched CD11c⁺ thymic cells from mixed bone marrow chimeras (50:50) of WT (CD45.1⁺) and *Cd11c^CreRosa26^TdTOMATO* (CD45.2⁺) mice. Mixed bone marrow chimeras (50:50) of WT (CD45.1⁺) and *Rosa26^TdTOMATO* (negative control; CD45.2⁺) mice were used as a negative control to TdTOM acquisition. (**c**) Quantification of CD45.1⁺TdTOM⁺CD11c⁺ cells from (**b**) (mean ± SEM, n = 11 mice from two independent experiments). Statistical analysis was performed using unpaired, two-tailed Student's *t*-test, ***p≤0.001. (**d**) Quantification of the frequency of TdTOM⁺ cells among the indicated dendritic cell (DC) subsets from reconstituted mice described in (**a**) (mean ± SEM, n = 11 mice from two independent experiments).

The online version of this article includes the following source data and figure supplement(s) for figure 5:

**Source data 1.** Thymic CD11c⁺ cells can share their antigens between each other.

**Figure supplement 1.** Thymic CD11c⁺ cells can share their antigens between each other.

APCs as a foundation, we have provided detailed insight into how particular subsets of TECs and thymic APCs are interconnected in the transfer of TEC-produced antigens. Specifically, utilizing several murine genetic models that allowed the tracking of TEC-produced antigen, we determined that CAT is mediated predominantly by preferential pairing between the following TEC and CD11c⁺ DC subsets: (i) cTECs to Mac and cDC2; (ii) mTEC^Low to Mac, cDC2, and pDC; (iii) mTEC^High to XCR1⁺ and XCR1⁻ aDC, and cDC1; (iv) pre-post Aire mTEC to XCR1⁺ and XCR1⁻ aDC; (v) post-Aire mTEC to pDCs and Mac; and (vi) Tuft mTEC to XCR1⁺ aDC. These CAT trajectories, which are depicted in *Figure 6a*, argue in favor of a model of preferential pairing in thymic antigen transfer. It seems that the antigen acquisition by Macs, cDC2s, and, to some extent, pDCs can occur when antigen is abundant. In addition, we also report that thymic moDCs, which do not exhibit subset specificity in CAT, generally obtain antigen from multiple cellular sources of thymic TECs as well as CD11c⁺ DC subsets.

In this study, we confirmed a high level of internal TEC heterogeneity that could be divided into a minimum of six distinct subsets (*Baran-Gale et al., 2020*; *Bautista et al., 2021*; *Bornstein et al., 2018*; *Dhalla et al., 2020*; *Wells et al., 2020*). Since the majority of these subsets are developmentally related (*Bornstein et al., 2018*; *Metzger et al., 2013*; *Miller et al., 2018*), our *Cre*-based *Rosa26^TdTOM* mouse models (*Figure 1a*) suggest some developmental relationships between mTEC subsets. It has been reported that TdTOM expression in the *Csnb^CreRosa26^TdTOM* mouse model is detected in a small proportion of mTEC^Low, in most mTEC^High, post-Aire mTEC, and Tuft mTEC subsets (*Bornstein et al.,*

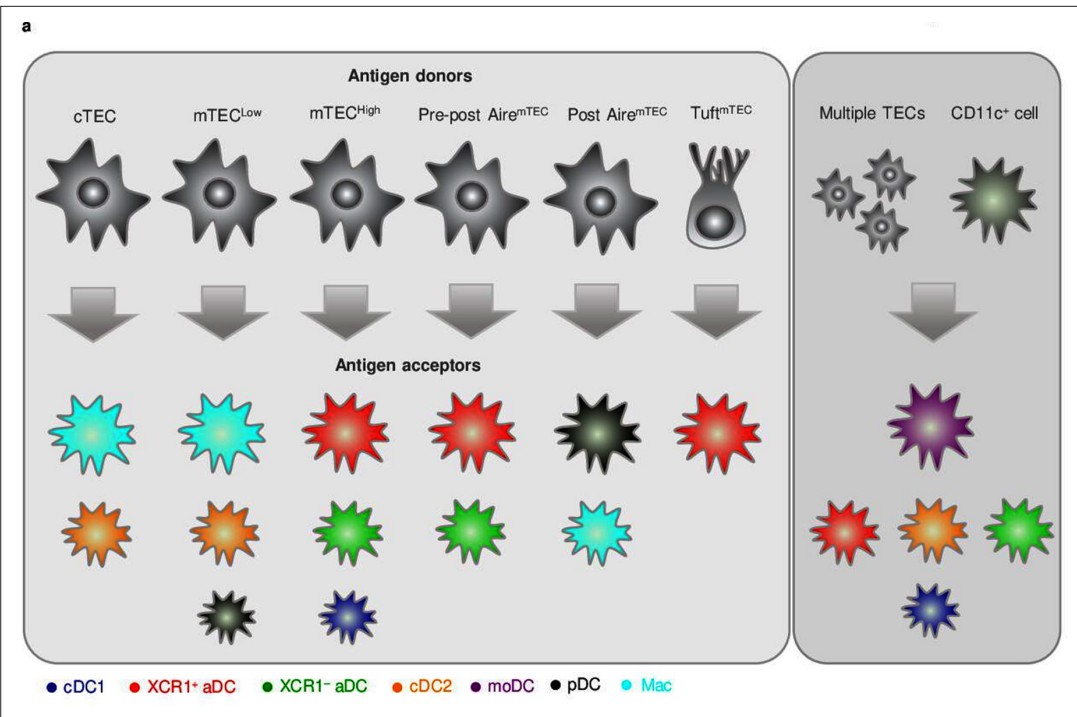

**Figure 6.** Proposed model of preferential pairing in cooperative antigen transfer (CAT). (**a**) Based on the data presented in this study, we postulate that phenotypically defined subsets of thymic CD11c⁺ cells preferentially and predictably acquire antigens from distinct subsets of developmentally related thymic epithelial cells (TECs) (left panel). Our data also suggests that thymic monocyte-derived dendritic cell (moDC), which represented the only CD11c⁺ subset that does not specifically prefer any particular subset of TECs (**Figure 3b**), excels in the acquisition of antigens from multiple TECs or other CD11c⁺ cells compared to both subsets of activated dendritic cell (aDC), cDC2, and cDC1, which also perform such modes of CAT, however, to a significantly lesser extent (right panel). Note that the size of the antigen acceptor's images reflects the strength of their correlation with particular TEC subset (left panel) or their potency in defined modes of CAT (right panel).

*2018*). This is consistent with our data (*Figure 1f*), which suggests that Csnb is expressed by a specific population of mTEC$^{Low}$ progenitors that further differentiate into mTECs$^{High}$ cells and later into their progeny. In contrast, the TdTOM expression in *Defa6$^{iCre}$Rosa26$^{TdTOM}$* should be specifically attributed to the Aire⁺ mTEC$^{High}$ subset and their post-Aire progeny since the expression of defensins in the thymus is highly dependent on Aire (*Dobeš et al., 2015*). Despite that, we also observed TdTOM expression in the small LY6D⁻ population of mTEC$^{Low}$ (*Figure 1f*). Since several distinct subpopulations of post-Aire mTEC were detected (*Dhalla et al., 2020*), we hypothesized that *Cre* recombination in mTEC$^{Low}$ likely reflects the presence of the LY6D⁻ population of post-Aire cells than *Defa6* locus activation in Aire⁻ mTEC$^{Low}$ progenitors. Thus, the significant correlation in CAT between mTEC$^{Low}$ and pDC subsets could be influenced by this phenomenon since pDCs were shown to acquire the antigen mostly from post-Aire mTECs (*Figure 3b*). It is also important to emphasize that all three mouse models used in this study were based on a *Cre* recombination system leading to the production of TdTOM in cells with an active *Foxn1*, or *Csn2*, or *Defa6* locus as well as to those cells that developmentally descend from these cells. Thus, the caveat of this approach is that the expression of TdTOM antigen is not targeted to a particular TEC subset exclusively, which would provide a more accurate system for assessing the preferential pairing of TEC and DC subsets in CAT.

The development of novel gating strategies has allowed us to reveal the substantial heterogeneity of thymic DCs that could be divided into phenotypically and functionally distinct subsets (*Li et al., 2009*; *Park et al., 2020*; *Vobořil et al., 2020*). Our data points to at least seven subtypes of CD11c⁺ cells that are capable of antigen acquisition from different subsets of TECs, that is, cDC1, XCR1⁺ aDC, XCR1⁻ aDC, cDC2, moDC, pDC, and Mac (*Figure 2e*, *Figure 2—figure supplement 2a and b*). Among them, we have phenotypically identified two subsets of thymic aDCs that are marked by the overexpression of the chemokine receptor, CCR7 (*Ardouin et al., 2016*; *Hu et al., 2017*; *Oh*

*et al., 2018*). Notably, it has been reported that the expression of *Ccr7* defines the population of XCR1+CCR7+ cDC1s that is considered to be the progeny of XCR1+CCR7− cDC1s (*Ardouin et al., 2016*). However, since these CCR7+ cDC1s express not only molecules that are associated with the cDC1 signature, such as *Batf3*, *Cd8α*, *Ly75*, or *Cadm1* (*Voboŕil et al., 2020*), but also molecules that characterize the population of aDCs (*Il12b, Il15, Il15ra Cd274, Cd70, Cd40, Tnfrsf4*) (*Ardouin et al., 2016*; *Park et al., 2020*), we defined and renamed this subset as XCR1+ aDC. Remarkably, these cells are the most efficient DC subset in CAT even when compared to cDC1s (*Figure 2—figure supplement 1b*, *Figure 4b*). It was recently suggested that the differentiation of XCR1+ aDCs from cDC1s is driven by the uptake of apoptotic cells (*Maier et al., 2020*). Since CAT has been shown to be mediated mostly by endocytosis of apoptotic bodies (*Koble and Kyewski, 2009*; *Perry et al., 2018*), the differentiation of XCR1+ aDCs in the thymus is consistent with being driven by CAT. Thus, the grounds for the correlation between mTEC^High and XCR1+ aDCs in TdTOM antigen transfer could be found in the fact that mTEC^High transfer antigen to XCR1+ cDC1 that further differentiate into XCR1+ aDC cells (*Ardouin et al., 2016*; *Maier et al., 2020*). In this context, it is also important to emphasize that the transcriptional signature of XCR1− aDCs is more similar to cDC2 (e.g., *Sirpa* and *Pdcd1lg2*) than the cDC1 subset (*Park et al., 2020*). By the same token, this suggests that antigen transfer into cDC2s induces their differentiation into XCR1− aDCs.

Using linear regression analysis of TdTOM+ TECs and DCs frequencies from all three mouse models, we identified two subsets of CD11c+ cells, cDC1 and moDC, that exhibited limited or no correlation with TEC subsets in TdTOM transfer. Surprisingly, the correlation of mTEC^High with cDC1s seems limited when compared to the correlation of mTEC^High with XCR1+ aDC (*Figure 3b*). This is contradictory to previous reports that described cDC1 subset as the most efficient in acquiring antigens from Aire+ mTEC^High (*Lei et al., 2011*; *Perry et al., 2018*). This could be explained by the fact that only XCR1 positivity was used by the authors to characterize cDC1 cells in the thymus. This does not allow one to discriminate between cDC1s and XCR1+ aDCs. Thus, as suggested by our data, the antigen transfer from mTEC^High is mediated mostly by XCR1+ aDCs and only to a lesser extent by cDC1s (*Figure 3b*). On the other hand, the linear regression model used in this study takes advantage of the variability in the ratios between TdTOM+ frequencies in TEC and DC subsets across all *Cre*-based *Rosa26^TdTOM* models. Since in the case of cDC1 this variability displays apparent limitations (*Figure 3a*), we see only inadequate correlation for this cell subset.

The second subset of thymic CD11c+ cells, which failed to show a correlation with any TEC subset in CAT, consisted of moDCs. Interestingly, while moDCs are very potent in CAT (*Figure 2e*, *Figure 2—figure supplement 1b*), their capacity can be further enhanced under inflammatory conditions (*Voboŕil et al., 2020*). We demonstrated that among other thymic DCs the moDC subset was the most efficient in repetitive CAT (*Figure 4b*, *Figure 5—figure supplement 1g*). This, along with their ability to efficiently acquire antigen from other CD11c+ APCs (*Figure 5d*), is a testament to their important function in central tolerance (*Park et al., 2020*; *Voboŕil et al., 2020*). Since thymic moDCs were shown to express a plethora of different chemokines and scavenger receptors (*Park et al., 2020*; *Voboŕil et al., 2020*), we propose that these characteristics correlate with their high competence in regulated migration and phagocytic activity compared to other DC subsets (*Croxford et al., 2015*).

In conclusion, using novel gating strategies for the identification of multiple TEC subsets that produce TdTOM antigen and tracking of its transfer into phenotypically defined thymic CD11c+ APC subsets has allowed us to define preferential antigen trajectories that mediate CAT. Our data shows that XCR1+ aDCs are the most potent subset in the acquisition of TEC-derived antigens. It also characterizes the moDC subset as the most efficient in the acquisition of antigen from multiple TECs as well as DCs. Taken together, our work proposes that CAT relies on a cellular interaction network with preferential partnerships between defined subtypes of TECs and DCs. This, in turn, suggests that the indirect presentation of antigens from developmentally related but phenotypically and functionally distinct types of TECs is ascribed to different subsets of thymic DCs. However, how these cell-to-cell preferential interactions, which are the underlying characteristics of CAT, facilitate the processes of central tolerance such as the deletion of self-reactive clones of T cells or their conversion to Tregs awaits resolution. Although this study suggests that CAT is a deterministic process, the molecules and mechanisms that determine TEC-to-DC cell-cell interactions remain to be identified.

# Materials and methods

## Mice

All mouse models used in this study were of C57BL/6J genetic background and housed under SPF conditions at the animal facility of the Institute of Molecular Genetics (IMG) in Prague. All animal experiments were approved by the ethical committee of the IMG and the Czech Academy of Sciences. C57BL/6J, *Foxn1^Cre^* (B6(Cg)-*Foxn1^tm3(cre)Nrm^*/J; MGI:5432012) (**Gordon et al., 2007**), Ly5.1 (B6.SJL-*Ptprc^a^ Pepc^b^*/BoyJ; MGI:2164701) (**Janowska-Wieczorek et al., 2001**), and *Cd11c^Cre^* (B6. Cg-Tg(Itgax-cre)1-1Reiz/J; MGI:3763375) (**Caton et al., 2007**) mice were purchased from The Jackson Laboratory. *Csnb^Cre^* mice (**Bornstein et al., 2018**) were kindly provided by J. Abramson (Department of Immunology, Weizmann Institute of Science, Rehovot, Israel). *Defa6^iCre^* mice (**Adolph et al., 2013**) were kindly provided by R. S. Blumberg (Division of Gastroenterology, Department of Medicine, Brigham and Women's Hospital, Harvard Medical School, Boston, MA). *Rosa26^TdTOMATO^* mice (B6;129S6-*Gt(ROSA)26Sor^tm14(CAG-TdTOMATO)Hze^*/J; MGI:3813512) (**Madisen et al., 2010**) were provided by V. Kořínek (IMG, Prague, Czech Republic). *Confetti^Brainbow2.1^* (*Gt(ROSA)26Sor^tm1(CAG-Brainbow2.1)Cle^*/J; MGI:4835546) (**Snippert et al., 2010**) mice were provided by the Czech Center for Phenogenomics (IMG, Vestec, Czech Republic). MHCII-EGFP mice-MGI:2387946 (**Boes et al., 2002**) were provided by J. Černý (Department of Cell Biology, Faculty of Science, Charles University, Prague). All mice were fed an Altromin 1314 IRR diet. Reverse osmosis filtered and chlorinated water was available to the animals ad libitum. All mice were bred in an environment in which the temperature and humidity of 22 ± 1°C and 55 ± 5%, respectively, were constant and under a 12 hr oscillating light/dark cycle. Prior to tissue isolation, mice were euthanized by cervical dislocation.

## Tissue preparation and cell isolation

Thymic tissue was extracted using forceps, cut into small pieces, and enzymatically digested with 0.1 mg*ml$^{-1}$ Dispase II (Gibco) dissolved in RPMI. Pieces of thymic tissue were pipetted up and down several times using a pipette tip that had been cut and incubated in a shaker at 800 rpm for 10 min at 37°C. This procedure was repeated approximately five times to completely dissolve the tissue. The supernatant was collected and the enzymatic reaction was stopped by adding 3% FCS and 2 mM EDTA. To isolate TECs, isolated cells were MACS-depleted of CD45$^+$ cells using CD45 microbeads (Miltenyi). After depletion, the suspension was spun down (4°C, 300 × *g*, 10 min) and the resulting pellet was resuspended in ACK lysis buffer for 2 min to deplete erythrocytes. To isolate thymic DCs and macrophages, MACS enrichment for CD11c$^+$ cells was performed using CD11c biotin-conjugated antibody (eBioscience) and Ultrapure Anti-Biotin microbeads (Miltenyi).

## Flow cytometry analysis and cell sorting

Cell staining for flow cytometry (FACS) analysis and sorting was performed at 4°C, in the dark, for 20–30 min, with the exception of anti-CCR7 antibody (BioLegend) staining that required incubation at 37°C for a minimum of 30 min. To exclude dead cells, either Hoechst 33258 (Sigma-Aldrich) or viability dye eFluor 506 (eBioscience) was used. FACS analysis of TECs and DCs was performed using BD LSR II and BD FACSymphony A5 cytometers, respectively. A BD FACSAria IIu sorter was used for cell sorting. BD FACSDiva software and FlowJo V10 software (Treestar) were used for FACS data analysis. For the purpose of tSNE analysis, the same amount of CD11c$^+$ TdTOM$^+$ cells from each model was concatenated by using the FlowJo concatenate function. The final tSNE was calculated by FlowJo opt–SNE plugin. Total TEC counts in *Figure 1e* were obtained by analyzing the whole fraction of CD45$^-$ MACS-enriched cells. Calculation of the ratios in *Figures 1g and 2g* is described in the 'Results' section and figure legends. The entire list of FACS staining reagents is provided in *Supplementary file 1*.

## Visualization of TdTOM expression in the thymus

To image TdTOM expression in *Foxn1^Cre^Rosa26^TdTOM^*, *Csnb^Cre^Rosa26^TdTOM^*, and *Defa6^iCre^Rosa26^TdTOM^* mice (*Figure 1d*), we extracted the thymi from these models as well as the *Cre$^-$* control mouse using forceps, submerged them in optimal cutting temperature solution, and froze on dry ice. The samples were stored overnight at –80°C. The next day, the thymi were cut into 10-μm-thick sections and put onto microscope slides. To stain thymic medullas, thymic sections were washed three times in PBS, fixed with 4% PFA in PBS for 10 min, washed again three times in PBS, permeabilized with 0.1% Triton X-100 in PBS for 15 min at room temperature (RT), and blocked for 30 min in 5% BSA and 0.1% Triton

X-100 in PBS (blocking buffer) at RT. Next, slides were incubated for 40 min with purified anti-keratin 14 antibody (BioLegend; clone: Poly19053) diluted 1:500 in blocking buffer, washed three times in PBS, and incubated for 20 min with goat anti-rabbit Alexa Fluor 647 secondary antibody (Life Technologies; Cat# A-21244) diluted 1:500 in blocking buffer. Before imaging, the samples were washed three times in PBS, gently washed in distilled water, and dried. Samples were imaged on Leica DM6000 widefield microscope using HC PL FLUOTAR ×10/0.30 DRY PH1; FWD 11.0; CG 0.17 | BF, DF, POL, PH objective. The tile scans of whole thymi were created using the LAS X Navigator tool (Leica). Importantly, bright-field and keratine 14 staining was imaged on sequential thymic sections to those used for TdTOM imaging. TdTOM was imaged right after cryosectioning.

### Imaging flow cytometry

Imaging flow cytometry (ImageStream) was performed using AMNIS ImageStream X MkII at the Center for Preclinical Imaging (CAPI) in Prague. Imaged XCR1$^+$ aDCs were isolated from *Foxn1$^{Cre}$-Confetti$^{Brainbow2.1}$* mice, stained for their CD11c, XCR1, and CCR7 markers, and sorted as RFP$^+$ and/or YFP$^+$ and/or CFP$^+$. The data was acquired via ImageStream with ×40 magnification. Ideas 6.1 software (AMNIS) was used for data analysis.

### Confocal and spinning disk microscopy

To test GFP expression in TECs from *Foxn1$^{Cre}$Confetti$^{Brainbow2.1}$* mice (**Figure 4—figure supplement 1a**), thymic cells from *Foxn1$^{Cre}$Confetti$^{Brainbow2.1}$* and MHCII-EGFP mice were MACS-depleted of CD45$^+$ fraction and imaged on a Leica TCS SP5 AOBS Tandem confocal microscope using the HCX PL APO ×10/0.40 DRY CS; FWD 2.2; CG 0.17 | BF, POL objective. To visualize TEC fluorescent variants from *Foxn1$^{Cre}$Confetti$^{Brainbow2.1}$* mice (**Figure 4—figure supplement 1d**), CD45$^+$EpCAM$^+$ TECs were sorted as RFP$^+$ and/or YFP$^+$ and/or CFP$^+$ and visualized in an uncoated μ-Slide 8 Well (ibidi) with Andor Dragonfly 503 spinning disk confocal microscope using HCX PL APO ×63/1.40–0.6 OIL $\lambda$ B; FWD 0.12; CG 0.17 | BF, POL, DIC objective.

### Bone marrow chimeras

BM was flushed out from the femur and tibia of Ly5.1 (CD45.1$^+$; **Figure 5**, **Figure 2—figure supplement 1d–f**, **Figure 5—figure supplement 1**) or *Cd11c$^{Cre}$Rosa26$^{TdTOMATO}$* (CD45.2$^+$; **Figure 5**, **Figure 5—figure supplement 1**) mice using a syringe with 26G needle. Isolated cells were depleted of erythrocytes with ACK lysis buffer. Recipient mice were sublethally irradiated with 6 Gy and reconstituted with 2 × 10$^6$ Ly5.1 BM cells in the case of *Foxn1$^{Cre}$/Csnb$^{Cre}$/Defa6$^{iCre}$Rosa26$^{TdTOMATO}$* mice (**Figure 2—figure supplement 1d–f**) or with 2 × 10$^6$, 50:50 mixed Ly5.1:*Cd11c$^{Cre}$Rosa26$^{TdTOMATO}$* BM cells in the case of C57BL/6J Ly5.1 mice (CD45.1$^+$CD45.2$^+$; **Figure 5**, **Figure 5—figure supplement 1**). Three weeks after irradiation, the BM reconstitution was verified by the staining of blood with anti-CD45.1 and CD45.2 antibodies. Mice were subjected to further analysis 6 weeks after irradiation if the BM reconstitution exceeded 80% (**Figure 2—figure supplement 1d–f**) or was between 40 and 60% within both CD45.1$^+$ and CD45.2$^+$ cell compartments (**Figure 5**, **Figure 5—figure supplement 1**).

### Statistical analysis

Statistical analysis and graph layouts were performed using Prism 5.04 software (GraphPad). Linear regressions were calculated using R 3.6.2. (R core team 2019). The statistical tests used for data analysis are indicated in the figure legends.

## Acknowledgements

We thank the members of the flow cytometry facility, Z Cimburek, and M Šíma, from IMG in Prague for technical support and I Novotný for assistance with microscopic experiments. We also thank M Báječný for technical assistance with Imaging flow cytometry. We are indebted to J Abramson for providing the *Csnb$^{Cre}$* mice, V Kořínek for *Rosa26$^{TdTOMATO}$* mice, and the Czech Center for Phenogenomics for *Confetti$^{Brainbow2.1}$* mice. This work was supported by Grant 20-30350 S from GACR. JB was supported by Grant 836119 from the Charles University Grant Agency (GA UK). JD was supported by a PRIMUS grant (Primus/21/MED/003) from Charles University and a Czech Science Foundation JUNIOR STAR grant (GAČR 21-22435 M). TB was partially supported by Grant RVO: 68378050-KAV-NPUI.

## Additional information

### Funding

| Funder | Grant reference number | Author |
|---|---|---|
| Czech Science Foundation | 20-30350S | Dominik Filipp |
| Charles University Grant Agency in Prague | 836119 | Jiří Březina |
| Charles University in Prague | Primus/21/MED/003 | Jan Dobeš |
| Czech Science Foundation | Junior Star grant 21-22435 | Jan Dobeš |
| Grant RVO Czech Republic | 68378050-KAV-NPUI | Tomáš Brabec |

The funders had no role in study design, data collection and interpretation, or the decision to submit the work for publication.

### Author contributions

Matouš Vo.bořil, Jiří Březina, Conceptualization, Data curation, Formal analysis, Investigation, Methodology, Visualization, Writing - original draft; Tomáš Brabec, Jan Dobeš, Formal analysis, Methodology; Ondřej Ballek, Formal analysis, Investigation, Visualization; Martina Dobešová, Investigation, Methodology, Resources; Jasper Manning, Investigation, Resources, Writing - review and editing; Richard S Blumberg, Resources; Dominik Filipp, Conceptualization, Data curation, Funding acquisition, Investigation, Methodology, Project administration, Resources, Supervision, Visualization, Writing - review and editing

### Author ORCIDs

Matouš Vo.bořil http://orcid.org/0000-0002-3672-7629
Jiří Březina http://orcid.org/0000-0001-6865-3874
Jan Dobeš http://orcid.org/0000-0003-1853-1603
Dominik Filipp http://orcid.org/0000-0003-2506-9330

### Ethics

All animal experiments in this study were performed in strict accordance with the protocol approved by the ethical committee of the Institute of Molecular Genetics and the Czech Academy of Sciences (Permit Number 27/2020 and 28/2020).

### Decision letter and Author response

Decision letter https://doi.org/10.7554/eLife.71578.sa1
Author response https://doi.org/10.7554/eLife.71578.sa2

## Additional files

### Supplementary files

• Supplementary file 1. List of antibodies.
• Transparent reporting form

### Data availability

All data generated or analyzed during this study are included in the manuscript and supporting files. Source data files have been provided for Figures 1, 2, 3, 4 and 5.

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
