## [Editor Report]

This manuscript will be of interest to immunologists studying mechanisms of thymic central tolerance. The study elegantly makes use of multiple genetic mouse models to generate data supporting the conclusion that different dendritic cell subsets in the thymus capture self-antigens from distinct subsets of thymic epithelial cells.

---

## [Decision Letter]

**Decision letter after peer review:**

Thank you for submitting your article "A model of preferential pairing between epithelial and dendritic cells in thymic antigen transfer" for consideration by *eLife*. Your article has been reviewed by 3 peer reviewers, including JC Zúñiga-Pflücker as Reviewing Editor and Reviewer #1, and the evaluation has been overseen by Tadatsugu Taniguchi as the Senior Editor.

Essential revisions:

1) The reviewers agreed that some additional analyses would strengthen the conclusions of the study. However, if these additional analyses lead to altered interpretation of the data presented then the authors should revise the model that was provided.

2) Please pay close attention to the detailed recommendations provided by Rev #3.

*Reviewer #1 (Recommendations for the authors):*

1- The main approach makes use of different mouse models expressing tdTomato under different cre-drivers, and it would be good to show immunofluorescence (IF) histological analysis of thymus section from these mice, which would further illustrate the elegance of the approach, by directly showing the expression pattern within the thymus medulla, or more broadly for the Foxn1-cre. Potentially, this analysis may even allow for the authors to show neighboring DCs that have picked up dtTomato as well.

2- For Figure 1, it could be useful to also show the cell numbers for the indicated subsets, rather than just the percentages.

*Reviewer #3 (Recommendations for the authors):*

Some additional analyses would strengthen the conclusions of the study, but they may lead to altered interpretation of the data presented and revision of the model. Here are some suggestions:

For Figure 1, to strengthen the conclusion about which TEC subsets are over-represented in the 3 reporter mice, it would be helpful to show the distribution of TEC subsets in the CD45-EpCAM+ gate (regardless of TdTomato). This is important because, for example, P5 line 122 indicates that "cTEC and mTEClow subsets were over-represented in Foxn1cre" reporter mice. However, comparison to the frequency of TEC subsets in Supplementary Figure 1 suggests that these TEC subsets are not over-represented in the reporter, but are instead represented according to their normal subset frequencies. Similar conclusions can be drawn for post-Aire TECs and Tuft cells in the Defa6cre reporter. Comparing the percent of TEC subsets within the Tomato+ gate to the percent of subsets overall would more accurately indicate if a given subsets is over- or under-represented. In addition, it would be advisable to show the % of TdTomato+ cells within each subset to help interpret negative results. Both of these analyses could be added to Figure 1, and results/conclusions in the text should be revised accordingly.

Similarly, in Figure 2g, the % of each DC subsets within the CD11c+Tomato+ cells is quantified, but this could be influenced by the frequency of different DC subsets at steady state and/or their ability to acquire TdTomato from different TEC subsets. So it would be helpful to the frequency of DC subsets out of total CD11c+ cells, relative to the frequency of each subset within the TdTomato+ CD11c gate for each reporter to more accurately determine if the DC subsets are truly over- or under-represented. To this point, P6 line 156 states, "XCR1+ aDCs were the most efficient cells involved in CAT irrespective of the Cre-based ROSA26TdTOM model used." However, the data show that ~40% of the tdTomato+CD11c+ cells are cDC1. From S2, ~ 40% of all Cd11c+ cells are cDC1. So instead of being more efficient at acquiring tdTomato from Foxn1cre reporter mTECs, they seem to be just the most abundant subset of DCs.

In Figure S2, showing the gating scheme for DC subsets, ~60% of the Xcr1-Ccr7- DCs do not express either MGL2 or CD14. This large subset of cDC2 cells seems to be excluded from downstream analyses. Do any of these cells acquire TdTomato in any of the reporters? Please either include the subset in the analyses or give a rationale for excluding them.

Figure 3a provides the overall summary of data in Figures 1 and 2, while Figure 3C presents the results of the linear regression between the frequencies of TdTomato TEC subset and TdTomato DC subsets based on the combined 3 reporters. This regression analysis is used to draw the model in Figure 6. However, some aspects of the model seem inconsistent with data in Figure 1. For example, given the reduced representation of mTEChi plus pre-post Aire+ mTEC in Defa6iCre (54%) relative to Csnbcre (68%) (Figure 3a), one would expect a reduction in the frequency of XCR1+ aDC in tomato+DCs according to the model in Figure 6, since this DC subset acquires antigen from both of these TEC subsets. However, the frequency of XCR1+aDC stays about the same between the two models, as seen by the insignificant difference in the % Xcr1+aDC within the CD11c+TdTOM+ compartment in Figure 1g. So it seems like the model may be overinterpreting the results. The authors acknowledge that the model isn't really accurately reflecting antigen acquisition, as it predicted that cDC1 interacted only with cTECs, despite evidence in the literature to the contrary (lines 224-235). Thus, the authors used an alternate regression model based on Csnbcre reporters at 2 different ages, which precited that cDC1 cells could acquire antigen from mTEChigh cells. However, the possibility should be considered that the frequency of the DC subsets changes between 4-6weeks and 11-13 weeks, so changes in the frequency of DC subsets in TdTomato+ could reflect that change, rather than antigen acquisition from mTEChigh cells. To test this the subset frequency of DCs at 4-6 weeks versus 11-13 weeks should be shown. Also, from the regression data in 3c, it seems like both XCR1+ and XCR1- aDCs can generally acquire antigen from mTEChi and Pre-post Aire cells, while cDC2 seem to acquire antigen from both Post Aire cells and mTEClow cells too. So if regression analysis is the basis for the model in Figure 6, it needs to be modified accordingly.

In Figure 4, only ~10% of RFP+ and/or YFP+ TECs in the Foxn1cre ConfettiBrainbow2.1 mice were cDC1. This is surprising given that cDC1s made up ~40% TdTomato+ DCs in Foxn1cre mice in Figure 3a. Have the authors stained for the different TEC populations in the rainbow confetti mice to see if RFP/CFP vs YFP expression is equally dispersed among the TEC subsets? If so, what accounts for this discrepancy?

In Figure 5, CD11c+ cell scavenging can occur from all cDC subsets, so the model in 6 should be revised to include these.

Given that this study does not confirm expression of Cre by specific TEC subsets, conclusions about lineage tracing using these models in the Discussion (p11-12 lines 315-337) should be softened significantly or eliminated. The data in the paper don't adequately justify the conclusions in this paragraph.

On page 12, 343-345, it is stated that the authors "defined two novel subsets of thymic aDCs, which are marked by the overexpression of the chemokine receptor CCR7." This is not novel, as Ardouin et al., defined these subsets within cDC1, as recognized by the authors, and Oh et al., (JI 2018; 200:1399-1412) and Hu et al., (Cell Reports 2017; 21:168-180) both showed expression of CCR7 by activated cDC1 and activated cDC2 subsets in the mouse thymus. Please modify the discussion accordingly.

---

## [Author Response]

Reviewer #1 (Recommendations for the authors):1- The main approach makes use of different mouse models expressing tdTomato under different cre-drivers, and it would be good to show immunofluorescence (IF) histological analysis of thymus section from these mice, which would further illustrate the elegance of the approach, by directly showing the expression pattern within the thymus medulla, or more broadly for the Foxn1-cre. Potentially, this analysis may even allow for the authors to show neighboring DCs that have picked up dtTomato as well.

We appreciate this suggestion. We have now included a novel set of data showing the histological analysis of TdTOM expression and localization in thymic tissue sections (see Figure 1d). This dataset is in the agreement with FACS data both of which point to dramatic quantitative differences in the TdTOM expression pattern among the mouse models used. Specifically, while TdTOM in Foxn1^Cre^ROSA^26TdTOM^ is expressed in whole thymic tissue, its expression in both the Csnβ^Cre^ROSA26^TdTOM^ and Defa6^iCre^ROSA26^TdTOM^ models is restricted to the medullary region of the thymus. In addition, the microscopic data from Defa6^iCre^ROSA26^TdTOM^ mice shows that the expression of TdTOM is restricted to a relatively small population of cells that reside in the thymic medulla. Since the Defa6 is a fully Aire-dependent TRA, this observation is consistent with the notion that the recombination of Defa6^iCre^ROSA26^TdTOM^ construct is microscopically detectable only in Aire^+^ mTECs^High^ and their descendants (see also the Results section and the discussion of the manuscript). On the other hand, this microscopic approach is not suitable for studying antigen transfer to thymic DCs, since it precludes the costaining of multiple DC-specific markers with TdTOM visualization of the same thymic slices (see the Materials and methods section). Additional transgenic mouse models would have to be created in future studies to overcome these technological obstacles.

2- For Figure 1, it could be useful to also show the cell numbers for the indicated subsets, rather than just the percentages.

Thank the Reviewer for this suggestion. We have now included a graph that compares the total numbers of tdTOM^+^ TECs between the models (see Figure 1e). We believe that this analysis complements the microscopic data from Figure 1d. On the other hand, the total number of individual TEC subsets in contrast to the ratios of the frequencies of TEC subsets within CD45^–^EpCAM^+^TdTOM^+^ cells to the frequencies of TEC subsets within CD45^–^EpCAM^+^ cells (see Figure 1g), can´t be used as a tool for studying the preferential pairing between TEC and DC subsets in CAT given that these numbers drastically decrease from Foxn1^Cre^ROSA^26TdTOM^ through Csnβ^Cre^ROSA26^TdTOM^ to Defa6^iCre^ models. Thus, we have decided to omit this type of information and instead we show only the ratios of normalized frequencies (Figure 1g).

Reviewer #3 (Recommendations for the authors):Some additional analyses would strengthen the conclusions of the study, but they may lead to altered interpretation of the data presented and revision of the model. Here are some suggestions:For Figure 1, to strengthen the conclusion about which TEC subsets are over-represented in the 3 reporter mice, it would be helpful to show the distribution of TEC subsets in the CD45-EpCAM+ gate (regardless of TdTomato). This is important because, for example, P5 line 122 indicates that "cTEC and mTEClow subsets were over-represented in Foxn1cre" reporter mice. However, comparison to the frequency of TEC subsets in Supplementary Figure 1 suggests that these TEC subsets are not over-represented in the reporter, but are instead represented according to their normal subset frequencies. Similar conclusions can be drawn for post-Aire TECs and Tuft cells in the Defa6cre reporter. Comparing the percent of TEC subsets within the Tomato+ gate to the percent of subsets overall would more accurately indicate if a given subsets is over- or under-represented. In addition, it would be advisable to show the % of TdTomato+ cells within each subset to help interpret negative results. Both of these analyses could be added to Figure 1, and results/conclusions in the text should be revised accordingly.

We thank the Reviewer for these suggestions. To show the differences in the distribution of TdTOM^+^ TECs between the models, we devised a new type of analysis. As suggested by the Reviewer, and for improving the overall transparency of our analysis, we now show the percentages of TEC subsets from TdTOM^+^ gate versus the percentages of these TEC subsets overall (Supplementary Figure 1c). These percentages were determined using the exploratory gating strategy illustrated in Supplementary Figure 1b. Then, the ratio between the percentage of a given TEC subset from TdTOM^+^ gate normalized to the percentage of the same TEC subset overall was calculated for each TEC subset. This ratio indicates the relative over- or underrepresentation of the former within the thymic CD45^–^EpCAM^+^TdTOM^+^ population among the three models used (Figure 1g). In this way, we replaced the plots of TEC subsets from TdTOM gates shown in Figure 1e in the original manuscript with the above described graphs with the ratios (Figure 1g). This analysis helped us to normalize the potential differences in the total frequency of TEC subsets between the models used caused by the litter effect, technical variability etc. This “normalization” allows one to report more accurately the increase/decrease in the frequency of TdTOM^+^ cells within the TEC-subset of interest. These ratios were then used for linear regression analysis visualized in Figure 3b. However, it is important to emphasize that trends in the distribution of TdTOM^+^ cells between mouse models are very similar when comparing the simple frequency of TdTOM^+^ TECs (used in our original manuscript) to the above-described ratios (normalized frequencies of TdTOM^+^ TECs). Also, based on the Reviewer´s suggestion, we have included the analysis of the frequency of TdTOM^+^ cells within the parent population (see Supplementary figure 1b).

Similarly, in Figure 2g, the % of each DC subsets within the CD11c+Tomato+ cells is quantified, but this could be influenced by the frequency of different DC subsets at steady state and/or their ability to acquire TdTomato from different TEC subsets. So it would be helpful to the frequency of DC subsets out of total CD11c+ cells, relative to the frequency of each subset within the TdTomato+ CD11c gate for each reporter to more accurately determine if the DC subsets are truly over- or under-represented. To this point, P6 line 156 states, "XCR1+ aDCs were the most efficient cells involved in CAT irrespective of the Cre-based ROSA26TdTOM model used." However, the data show that ~40% of the tdTomato+CD11c+ cells are cDC1. From S2, ~ 40% of all Cd11c+ cells are cDC1. So instead of being more efficient at acquiring tdTomato from Foxn1cre reporter mTECs, they seem to be just the most abundant subset of DCs.

We thank the reviewer for this constructive input. To address this issue, we implemented the same strategy as described above for TEC subsets. Notably, we calculated the percentages of DC subsets from TdTOM^+^ gate (TdTOM^+^ DCs) versus the same DC subsets overall (All DCs) which in the revised version of the manuscript are presented in Supplementary Figure 3c. From these frequencies we calculated their ratios, analogously as in the Figure 1g for TECs (Figure 2g). Finally, we used the TEC and DC ratios in the linear regression analysis, the results of which are preseted in Figure 3b and c.

As far as the concern regarding our statement that "XCR1^+^ aDCs were the most efficient cells involved in CAT irrespective of the Cre-based ROSA26TdTOM model used”, we believe that this is a misunderstanding. The reason being that XCR1^+^ aDCs are not the most abundant subset of DCs in the thymus (see Supplementary figure 2a). Quite contrary, they are rather underrepresented among the DC-subsets (see Author response image 1). In general, given their relatively low frequency, this makes our statement that XCR1^+^ aDCs are the most efficient population in CAT even more convincing and stronger.

**Author response image 1. sa2fig1:** 

In Figure S2, showing the gating scheme for DC subsets, ~60% of the Xcr1-Ccr7- DCs do not express either MGL2 or CD14. This large subset of cDC2 cells seems to be excluded from downstream analyses. Do any of these cells acquire TdTomato in any of the reporters? Please either include the subset in the analyses or give a rationale for excluding them.

This is an interesting point. As pointed out by the Reviewer, in the TdTOM-independent gating strategy (Supplementary figure 2a), we see around 60% of XCR1^-^CCR7^-^ that do not express either MGL2 or CD14. On the other hand, this population is almost negligible when gating on TdTOM^+^ CD11c^+^ cells (see Supplementary figure 3b). Thus, this suggests that XCR1^-^CCR7^-^MGL2^-^CD14^-^ cells have a very limited capacity in acquiring TdTOM in any of the reporters. Together with the fact that these cells could not be defined by the expression of established DC markers, argues that these cells could be of mixed origins, which led us to exclude them from our downstream analysis.

Figure 3a provides the overall summary of data in Figures 1 and 2, while Figure 3C presents the results of the linear regression between the frequencies of TdTomato TEC subset and TdTomato DC subsets based on the combined 3 reporters. This regression analysis is used to draw the model in Figure 6. However, some aspects of the model seem inconsistent with data in Figure 1. For example, given the reduced representation of mTEChi plus pre-post Aire+ mTEC in Defa6iCre (54%) relative to Csnbcre (68%) (Figure 3a), one would expect a reduction in the frequency of XCR1+ aDC in tomato+DCs according to the model in Figure 6, since this DC subset acquires antigen from both of these TEC subsets. However, the frequency of XCR1+aDC stays about the same between the two models, as seen by the insignificant difference in the % Xcr1+aDC within the CD11c+TdTOM+ compartment in Figure 1g. So it seems like the model may be overinterpreting the results. The authors acknowledge that the model isn't really accurately reflecting antigen acquisition, as it predicted that cDC1 interacted only with cTECs, despite evidence in the literature to the contrary (lines 224-235). Thus, the authors used an alternate regression model based on Csnbcre reporters at 2 different ages, which precited that cDC1 cells could acquire antigen from mTEChigh cells. However, the possibility should be considered that the frequency of the DC subsets changes between 4-6weeks and 11-13 weeks, so changes in the frequency of DC subsets in TdTomato+ could reflect that change, rather than antigen acquisition from mTEChigh cells. To test this the subset frequency of DCs at 4-6 weeks versus 11-13 weeks should be shown. Also, from the regression data in 3c, it seems like both XCR1+ and XCR1- aDCs can generally acquire antigen from mTEChi and Pre-post Aire cells, while cDC2 seem to acquire antigen from both Post Aire cells and mTEClow cells too. So if regression analysis is the basis for the model in Figure 6, it needs to be modified accordingly.

We thank the Reviewer for this important insight. Based on the Reviewer’s suggestions, Figure 3 has been considerably revised (see updated Figure 3). In brief, we used the above described ratios to reveal a potential preferential pairing between the TdTOM^+^ cells in TECs and thymic DCs. The overall summary (Figure 3a) now shows the ratios of the frequency of each TEC and DC subset from the dTOM^+^ gate to the frequency of a particular population overall across the models used. The dashed line indicates which populations in TECs and DCs are over/underrepresented in a particular mouse model. Then, these ratios were used for linear regression analysis between the TEC and DC subsets (see Figure 3b and c). Since using normalized frequencies of TdTOM^+^ cells for calculating the ratio values for each of these subsets slightly changes the output of this regression analysis, the model of preferential pairing in CAT between the TECs a DCs presented in Figure 6 has been modified accordingly.

Also, based on the Reviewer's comments we decided to omit the analysis comparing the frequencies of TdTOM^+^ cells between younger and older Csnβ^Cre^ mice. We agree with the reviewer, that there exist age-related changes in the composition of thymic DC compartment, the phenomenon which has been already described in the literature (Kroger JCH., et al., European Journal of Immunology 2016, Van Dommelen SLH., et al., Cellular and Molecular Immunology 2010). Although these changes can be normalized by using the ratios as in Figure 3a, we realized that many factors such as decreasing T cells input into the thymus or overall involution of the thymic tissue might also affect the function/state of DCs including their capacity to perform CAT in older Csnb^Cre^ mice (Baran-Gale J., et al., *eLife* 2020, Mittelbrunn M. and Kroemer G., Nature Immunology 2021). By analysing only the “young” models, we thus avoided the incorporation of these and other variables, which might have led to misrepresentation of this dataset.

In Figure 4, only ~10% of RFP+ and/or YFP+ TECs in the Foxn1cre ConfettiBrainbow2.1 mice were cDC1. This is surprising given that cDC1s made up ~40% TdTomato+ DCs in Foxn1cre mice in Figure 3a. Have the authors stained for the different TEC populations in the rainbow confetti mice to see if RFP/CFP vs YFP expression is equally dispersed among the TEC subsets? If so, what accounts for this discrepancy?

We thank the Reviewer for pointing out this discrepancy. We stained for some of the TEC populations described in this study and could not detect any changes in RFP/CFP or YFP expression between the subsets analyzed (see Author response image 2). First, it is important to emphasize that Figure 4 compares only the frequencies of fluorescent protein^+^ cells within the parent population which is different from the comparisons shown in Figure 2. Also, compared to the expression of TdTOM under the Foxn1^Cre^ system (Supplementary Figure 2b), different fluorescent proteins in Foxn1^Cre^Confetti^Brainbow2.1^ model are not expressed in the same concentration by all TECs as TdTOM. Notably, around 25% of TECs which were supposed to express GFP don´t express any fluorescent protein (Supplementary figure 4b and c). Moreover, graphs depicted in Figure 4 compare the antigen transfer of only RFP and/or YFP and not RFP^+^YFP^+^ (shown in separate graph) excluding all DCs that performed the “double transfer” from the analysis of Foxn1^Cre^Confetti^Brainbow2.1^ model but those were not (and could not be) excluded in the case of Foxn1^Cre^ROSA^26TdTOM^ model analysis (Supplementary Figure 2b). Finally, around 30% of TECs express the membrane-associated CFP antigen whose transfer is much more restricted compared to the one of cytoplasmic TdTOM. Thus, due to these considerations, we believe that the comparison of CAT between Foxn1^Cre^Confetti^Brainbow2.1^ and Foxn1^Cre^ROSA^26TdTOM^ models would not be appropriate.

In Figure 5, CD11c+ cell scavenging can occur from all cDC subsets, so the model in 6 should be revised to include these.

This point is well taken. We have modified the model in Figure 6 accordingly.

Given that this study does not confirm expression of Cre by specific TEC subsets, conclusions about lineage tracing using these models in the Discussion (p11-12 lines 315-337) should be softened significantly or eliminated. The data in the paper don't adequately justify the conclusions in this paragraph.

We thank the Reviewer for this point. The discussion has been modified and the tone regarding lineage tracing softened.

On page 12, 343-345, it is stated that the authors "defined two novel subsets of thymic aDCs, which are marked by the overexpression of the chemokine receptor CCR7." This is not novel, as Ardouin et al., defined these subsets within cDC1, as recognized by the authors, and Oh et al., (JI 2018; 200:1399-1412) and Hu et al., (Cell Reports 2017; 21:168-180) both showed expression of CCR7 by activated cDC1 and activated cDC2 subsets in the mouse thymus. Please modify the discussion accordingly.

We thank the Reviewer for raising this issue. We apologize for not including these citations in our manuscript. We have modified this section of the manuscript accordingly and the references have been included in the manuscript.